# G-Quadruplex Regulation of *VEGFA* mRNA Translation by RBM4

**DOI:** 10.3390/ijms23020743

**Published:** 2022-01-11

**Authors:** Kangkang Niu, Xiaojuan Zhang, Qisheng Song, Qili Feng

**Affiliations:** 1Guangdong Provincial Key Laboratory of Insect Developmental Biology and Applied Technology, Institute of Insect Science and Technology, School of Life Sciences, South China Normal University, Guangzhou 510631, China; kkniu@m.scnu.edu.cn (K.N.); Zhangxj@m.scnu.edu.cn (X.Z.); 2Guangzhou Key Laboratory of Insect Development Regulation and Application Research, Institute of Insect Science and Technology, School of Life Sciences, South China Normal University, Guangzhou 510631, China; 3Division of Plant Sciences, University of Missouri, Columbia, MO 65211, USA; songq@missouri.edu

**Keywords:** G-quadruplex, *VEGFA*, IRES, mRNA translation

## Abstract

In eukaryotes, mRNAs translation is mainly mediated in a cap-dependent or cap-independent manner. The latter is primarily initiated at the internal ribosome entry site (IRES) in the 5′-UTR of mRNAs. It has been reported that the G-quadruplex structure (G4) in the IRES elements could regulate the IRES activity. We previously confirmed RBM4 (also known as LARK) as a G4-binding protein in human. In this study, to investigate whether RBM4 is involved in the regulation of the IRES activity by binding with the G4 structure within the IRES element, the IRES-A element in the 5′-UTR of vascular endothelial growth factor A (*VEGFA*) was constructed into a dicistronic reporter vector, psiCHECK2, and the effect of RBM4 on the IRES activity was tested in 293T cells. The results showed that the IRES insertion significantly increased the FLuc expression activity, indicating that this G4-containing IRES was active in 293T cells. When the G4 structure in the IRES was disrupted by base mutation, the IRES activity was significantly decreased. The IRES activity was notably increased when the cells were treated with G4 stabilizer PDS. EMSA results showed that RBM4 specifically bound the G4 structure in the IRES element. The knockdown of RBM4 substantially reduced the IRES activity, whereas over-expressing RBM4 increased the IRES activity. Taking all results together, we demonstrated that RBM4 promoted the mRNA translation of *VEGFA* gene by binding to the G4 structure in the IRES.

## 1. Introduction

In protein synthesis, the genetic information in mRNA is translated into an amino acid sequence with the help of ribosomes and transfer RNAs (tRNAs) [1]. The mRNA translation is mediated in either a cap-dependent or cap-independent manner [2,3]. The former is a main regulation mode of mRNA translation in eukaryotes. However, when cells are suffering environmental stress, such as heat shock, hypoxia, amino acid deficiency, and viral infection, the cellular responses immediately lead to global repression of cap-dependent protein synthesis. Meanwhile, alternative mechanisms are activated to support the translation of specific mRNAs [4,5].

The mRNA translation can be subjectively separated into three stages: initiation, extension, and termination. The main difference between cap-dependent and cap-independent translation lies in the initiation stage. The initial step of cap-dependent translation initiation involves the binding of ribosomes, together with tRNAMet, to the 7-methylguanosine (m7-GpppG) 5′-cap of an mRNA. However, some viral and cellular mRNAs contain an internal ribosome entry site (IRES) that attracts ribosomes directly to the interior of the RNA to initiate cap-independent translation, which is also known as IRES mediated translation. IRES-mediated translation was first found in the RNAs of the virus [6,7]. Virus mRNA are uncapped and translated in a cap-independent manner. Eukaryotic cellular IRES elements were also identified, especially in the 5′-UTR of mRNA of some proto-oncogenes [8,9,10]. When global translation is blocked in response to stress to save energy, these genes were translated in the IRES dependent way to synthesize necessary proteins [3,11]. The mRNA IRES sequence can form a variety of RNA structures, such as stem loop, hairpin, G4 structures, and other secondary or tertiary structures. These advanced RNA structures can replace the 5′-cap of mRNA and recruit ribosomes to initiate the translation process [12,13]. The G4 structure is a special secondary structure of DNA and RNA. When nucleic acid sequences enrich in guanine (G), these sequences are favorable to form G planes, which consist of four Gs linked together through Hoogsteen hydrogen bond, and two or more G planes are stacked to form the G4 structure [14,15,16,17]. The G4 structure in the IRES sequence was found to directly bind with the 40S small subunit of ribosome, implying its regulatory role in the IRES-mediated translation [18].

The human vascular endothelial growth factor (VEGF) is a homodimeric glyco protein which plays critical roles in vasculogenesis and angiogenesis. The VEGF family consists VEGF-A (hereafter called VEGFA), VEGF-B, VEGF-C, VEGF-D, VEGF-E (viral VEGF), VEGF-F (snake venom VEGF), placenta growth factor (PlGF), and the endocrine gland-derived vascular endothelial growth factor (EG-VEGF). More attention was focused on the VEGFA due to its key roles in tumor growth and metastasis and eye disease [19]. The 5′UTR of *VEGFA* is G-rich and harbors two separate IRES sites (IRES-A at 749–1038 nt and IRES-B at 91–554 nt) that facilitate *VEGFA* translation initiation in a cap-independent manner to respond to various cellular stresses [20,21]. It has been reported that IRES-A could initiate *VEGFA* translation at AUG and synthesize a secreted form of VEGFA, which is crucial for the tumor angiogenesis [22]. Moreover, a G4 structure was identified in IRES-A region and the G4 structure is needed for *VEGFA* IRES mediated translation [23]. Consequently, the IRES-A of *VEGFA* was used to investigate the specific mechanism of G4 structures in regulating mRNA translation.

Human RNA-binding motif protein 4 (RBM4), a homologue of LARK in *Bombyx mori*, is ubiquitously expressed in human tissues [24]. We previously demonstrated that LARK or RBM4 is a G4-binding protein in silkworms, mice, and human [25]. Studies have shown that RBM4 is mainly involved in biological processes, such as selective RNA splicing and RNA translation [26,27]. RBM4 has been reported to inhibit cap-dependent translation but promote the IRES-mediated translation under cellular stress [24,26]. However, how RBM4, as a G4-binding protein, participates in IRES-mediated translation remains to be elucidated.

This study takes IRES-A of *VEGFA* as a target gene to explore whether RBM4 can promote the IRES-mediated mRNA translation by binding the G4 structure in the IRES element. The results revealed that RBM4 favors IRES-mediated translation as a G4 binding protein.

## 2. Results

### 2.1. IRES-A Promoted the VEGFA mRNA Translation in 293T Cells

To investigate whether the IRES-A element of *VEGFA* 5′-UTR mediates mRNA translation, the dual luciferase reporter psiCHECK2 was applied to detect its function in mRNA translation. The HSV-TK promoter upstream of the firefly luciferase (FLuc) marker gene was replaced by the *VEGFA* IRES-A sequence to generate a pR-IRES-F vector. The control pRF was the empty vector without the HSV-TK promoter (Figure 1A). The pR-IRES-F plasmid was used to transfect the 293T cells, followed by a luciferase activity assay after 24 h. The results showed that *VEGFA* IRES-A activity (FLuc/RLuc) was significantly increased, as compared to the control plasmid pRF (Figure 1B). Further analysis showed that the RLuc activity was not significantly changed, while FLuc activity was significantly increased (Figure 1C). These results suggest that the *VEGFA* IRES-A could promote either the transcription or translation of FLuc in the 293T cells.

Various mechanisms could account for the increase of Fluc translation downstream of the dicistronic mRNA. The dicistronic mRNA may be spliced into monocistronic mRNAs. The inserted IRES sequence may contain an element that promotes the read-through past the RLuc cistron and causes reinitiation of Fluc cistron. Finally, the inserted sequence may act as an IRES site mediating cap-independent translation. In order to determine whether the effect of the *VEGFA* IRES-A on the increase of the FLuc activity occurs at the mRNA translation level or transcriptional level, and to further verify that RLuc and FLuc are translated from the same mRNA, the upstream primer and downstream primer within the RLuc and FLuc genes respectively were specifically designed to detect whether the mRNA was spliced and the mRNA level of individual genes was examined. Total RNA were extracted from the 293T cells transfected with pRF and PR-IRES-F plasmids and reverse transcribed into cDNA. PCR was then performed to detect whether mRNA is spliced between the *RLuc* and *FLuc* mRNA. The results showed that the resultant PCR band was a single one, the size was expected, and there was no cleavage between *RLuc* and *FLuc* mRNAs in both the pRF and PR-IRES-F cases (Figure 1D). In addition, the qPCR results showed that there was no difference in mRNA expression between *RLuc* and *FLuc* in both the pRF and PR-IRES-F vectors (Figure 1E). These results indicated that the protein expression (as indicated by luciferase activity) of RLuc and FLuc was the result of the translation of the same mRNA without splicing.

Besides, to verify that the FLuc translation was not caused by mRNA read-through during RLuc translation, we added a hairpin structure upstream of the *RLuc* sequence (Figure 1A). This hairpin structure significantly inhibited the RLuc activity, whereas the FLuc activity was not impacted (Figure 1F). If the insertion could enhance ribosomal read-through, the Fluc activity should be reduced by an equivalent amount. Thus, given that the promoter (SV40) of the two vector constructs was the same, the significant increase in the FLuc enzyme activity was considered to be the result of the increased IRES mediated translation.

### 2.2. IRES Activity Was Affected by G4 Structure

It has been demonstrated by DNAase I footprinting that the 774~90 nt region of the IRES-A sequence in the *VEGFA* 5′-UTR could form a G4 structure and this G4 structure had a regulatory effect on the IRES activity [23]. In this study, the CD experiments revealed that the oligonucleotide of this mRNA region had a maximum absorption peak at 264 nm, and the peak was increased when 100 mM K^+^ was added (Figure 2A), presenting a classical peak type of parallel G4 structure [28,29]. When the G4 sequence was mutated, the maximum absorption peak shifted to 275 nm, and the K^+^-dependent increase of the absorption peak disappeared (Figure 2A), indicating that the mutant RNA could not form G4 structures any more consistent than previously reported [28,29]. The G4 mutation resulted in a significant decrease in the FLuc activity, but not in the RLuc activity (Figure 2B). Moreover, when the cells were treated with 2.5 μM PDS, a small molecular compound known to bind and stabilize the G4 structure, the FLuc activity was significantly increased, as compared to the control (DMSO), while the Rluc activity was not significantly changed (Figure 2C). When the G4 sequence was mutated, the FLuc activity did not respond to the PDS treatment (Figure 2D). These results suggested that the IRES-A element could form a G4 structure that facilitated the IRES-mediated FLuc mRNA translation.

### 2.3. RBM4 Specifically Bound the G4 Structure in IRES Element

To verify whether RBM4 binds the G4 structure in the *VEGFA* IRES-A, RNA G4 oligonucleotides were synthesized in vitro and annealed to allow the formation of the G4 structure, followed by EMSA with purified RBM4 protein. The result showed that RBM4 could specifically bind the G4 structure (Figure 3A, lane 2) and this specific binding could be competed off gradually by the unlabeled probe (Figure 3A, lane 3 and 4). It is noted that there were four binding bands, implying that various G4 structure isforms or formats of the interaction between the G4 structure and the protein might be present in the binding system. The mutated RNA G4 probe could not bind the RBM4 protein (Figure 3A, lane 5). When the protein content of RBM4 was increased gradually, the binding intensity was strengthened (Figure 3B). These results indicated that RBM4 could specifically bind the G4 structure in the *VEGFA* IRES-A element.

### 2.4. RBM4 Promoted the VEGFA IRES-A Activity

To investigate the effect of RBM4 on the IRES-A translation activity, we conducted RBM4 RNAi and over-expression in the 293T cells. The FLuc enzyme activity was significantly decreased when the RBM4 was down-regulated, as compared to that of the control (Figure 4A,B). The RBM4 RNAi did not affect the FLuc activity when the G4 sequence was mutated (Figure 4B). Contrarily, the FLuc activity was notably increased when RBM4 protein was over-expressed (Figure 4C,D). Similarly, the FLuc enzyme activity did not change when the G4 sequence was mutated (Figure 4D). Taken together, these results suggest that RBM4 could specifically bind the G4 structure in the *VEGFA* IRES-A and enhance the IRES-mediated translation of *VEGFA* mRNA.

## 3. Discussion

IRES sequences were firstly found in viral genomes and subsequently identified in eukaryotic cells [30]. When cells suffer cellular stress, such as heat shock, hypoxia, and viral infection, the classic cap-dependent translation is inhibited, blocking protein synthesis. However, the stress-induced attenuation of global translation is complementarily accompanied by selective translation of mRNAs that possess IRES [26]. The IRES translation activity has been detected in different types of cells. For example, the IRESes of *c-MYC* and *FGF-2* genes were active in developing embryos [31,32]. The IRES of *Apaf-1* gene showed higher IRES activity in neurogenic cell lines than in HEK293, MCF7, MRC5 and Cos7 cell lines [33]. In this study, we demonstrated that the *VEGFA* IRES-A element was active in the regulation of mRNA translation in the 293T cell line (Figure 1B). Although vast numbers of translational regulating IRESes were identified in viral and cellular genes, the mechanism of IRES mediated translation remains unclear. There was no similarity among the reported IRES sequences. Interestingly, IRES sequences can form various types of secondary structures, such as stem loop, hairpin, and G4 structures, which could promote the corresponding mRNA translation by recruiting translation initiation factors [13]. The ribosome 40S small subunit can bind the G4 structure in mRNA to promote IRES-mediated translation [18]. Here, we demonstrated that the G4 structure within IRES of *VEGFA* 5′UTR was needed to activate the cap-independent translation (Figure 5).

The G4 structure is a four-stranded secondary structure of nucleic acid. About 700,000 G4 structures have been predicted in the human genome, and most of these G4 structures are distributed in the promoter and UTR regions of genes [34]. G4 has been reported to be involved in a variety of biological processes, such as gene transcription, DNA replication, telomere protection, RNA splicing, and translation [35]. Compared to DNA G4, RNA G4 structures are more stable, diverse, and most of them could form a parallel G4 structure [36]. RNA G4s can act as either an obstacle impeding ribosome scanning to inhibit translation or a recognition site for ribosomes or splicing factors to promote translation and splicing [36]. A G4 formation sequence in the IRES-A of VEGFA 5′UTR was identified and revealed to be benefit IRES-mediated translation [23]. In this study, CD analysis verified the formation of a parallel G4 structure, which is consistent with the usual characteristics of RNA G4s. The treatment with small molecule G4-binding compound PDS resulted in an increase in the IRES translation activity (Figure 2C), suggesting that a stable G4 structure facilitates the IRES translation activity. In addition, G4 sequences in 5′UTR of *VEGFA* were conserved among 10 mammal species (Appendix A), implying that the translational regulation mechanism of *VEGFA* may be evolutionarily conserved in mammals.

RBM4 protein is an RNA binding protein that contains two classical RNA recognization motifs (RRM) and one zinc finger domain. RBM4, a homolog of silkworm LARK, was first identified in *Drosophila melanogaster*, and the LARK mutant resulted in early eclosion [37]. We previously found that LARK can bind the G4 structure in the promoter region of a variety of genes, as well as the RNA G4 structure [24]. Human RBM4 protein is highly homologous to the LARK protein of silkworm and *Drosophila* [25], implying RBM4 may also be a G4 stabilizer or inducer. RBM4 protein can bind to the G4 structure in the promoters of *MYC* and *KIT* gene [25]. In this study, we found that RBM4 can specifically bind the *VEGFA* mRNA G4 structure promoting the translation activity of the mRNA. RBM4 is phosphorylated in cytoplasm when cells encounter cellular stress, and then interacts with translation initiation factor eIF4A to promote the translation activity of the IRES [26]. Our study demonstrated that RBM4 action in promoting the IRES-mediated mRNA translation was achieved by binding with and stabilizing the G4 structure in the IRES, emphasizing the critical role of the G4 structure in mRNA translation. We hypothesize that G4 structures formed in the IRES element recruit its binding protein RBM4 and interact with 40S ribosome subunit. Then, RBM4 interacts with translation initiation factor eIF4A, forming the translation initiation complex, enhancing the cap-independent translation in stress condition (Figure 5).

VEGFA is a key mediator of angiogenesis in cancer. VEGFA is overexpressed in tumor cells, and VEGFA protein secreted by tumor cells and surrounding stroma induces the formation of new blood vessels, leading to tumor outgrowth, metastasis, and prognosis [38,39]. Given the critical role of VEGFA in tumor angiogenesis and a perfect anticancer target, the translational regulation of VEGFA has received growing attention. In tumor tissues, fast-growing cells encounter hypoxic environment. Hypoxia represses most mRNA translation, while VEGFA could be translated in a IRES dependent way, promoting the angiogenesis. If the translation of *VEGFA* could be affected, tumor growth and metastasis could be inhibited to some extent. Here, we demonstrated that the G4 structure at *VEGFA* IRES-A facilitated its cap-independent translation through binding with RBM4, elaborating the mechanism whereby the G4 structure in the IRES element affects the IRES-mediated translation. Compounds which could specifically target the G4 structure in *VEGFA* IRES, affecting its folding or unfolding, and then affecting VEGFA expression, would be novel potential therapeutic applications. As a matter of fact, a quinazoline-based compound that targets G4 structures has been revealed to inhibit DNA replication, DNA damage checkpoint activation, and promote apoptosis [40]. Moreover, RBM4, known as an RNA binding protein here, was verified to be involved in cap-independent translation as G4 binding protein, broadening our understanding of RBM4 and guiding us to investigate the function of RBM4 on carcinogenesis.

## 4. Materials and Methods

### 4.1. Vector Construction

The 306 bp *VEGFA* (Gene ID: 7422) IRES-A sequence (with Xho I and Mlu I restriction enzyme sites at 5′ and 3′ respectively) obtained from IRESbase, available online: http://reprod.njmu.edu.cn/cgi-bin/iresbase/index.php website (accessed on 5 January 2022) was synthesized and inserted into pMD18T vector (Qingke, Beijing, China). The dual luciferase reporter vector psiCHECK2 was used to study the IRES activity. The HSV-TK promoter region was excised with unique Xho I and Mlu I restriction sites, and replaced with the VEGFA IRES-A fragment, generating a pR-IRES-F vector. The control vector pRF was obtained by self-ligation of the blunted liner psiCHECK2 vector digested by Xho I and Mlu I. To exclude ribosomal read-through, a 60 bp oligonucleotides encoding a stable hairpin was cloned into the unique Nhe I site upstream of the Renilla luciferase, resulting in a p-hp-R-IRES-F vector. The sequence of the hairpin was as follows: 5′-AGCCATGGTGCTAGAGAGATCTGGTACCGAGCTCCCCGGGCTGCAGGATATCCTGCAGCCCGGGGAGCTCGGTACCAGATCTAGCCTATAGTGAG-3′.

To construct the RBM4 eukaryotic expression vector, the open reading frame (ORF) sequence of RBM4 was cloned with cDNAs of 293T cells as template. Specific primers for cloning RBM4 ORF were as follows: Forward primer: 5′-GGTACCGCCACCATGGTGAAGCTGTTCATCGGAA-3′; Reverse primer: 5′-GGATCCAAAGGCTGAGTACCGCGC-3′. The RBM4 ORF fragment was cloned into the EGFP-N1 vector with Kpn I and BamH I, generating a RBM4-GFP vector.

### 4.2. Cell Culture and Transfection

The 293T cell line was purchased from ATCC. The cells were cultured at 37 °C in an incubator containing 5% CO_2_ with DEME medium supplemented with 10% fetal bovine serum (FBS). The cells were passaged every 2–3 days.

The 293T cells were inoculated in 12-well cell culture plates prior to transfection. Transfection was conducted when the cells grew to 80–90% density the next day. A 50 μL mixture containing 1 μg plasmid DNA (pRF or PR-IRES-F), 3 μL Fugene liposome, and Opti-MEM medium was added to the medium after 15 min incubation at room temperature. Then, 24 h later, the cells were collected, and the enzyme activity was determined. For the over-expression of RBM4, 1 μg PR-IRES-F vector DNA and 1 μg RBM4-GFP over-expression vector DNA (control EGFP-N1 vector) were transfected into the cells. For RNA interference (RNAi), 1 μg PR-IRES-F vector DNA and 40 pmol RBM4 siRNA (non-specific siRNA as control) were co-transfected into the cells. The cells were collected at 24 h post transfection and the enzyme activity was measured.

### 4.3. Dual Luciferase Assay

Luciferase activity was measured using the Dual Luciferase Kit (Yishengbio, Shanghai, China). The treated cells were washed once with PBS and then lysed with 300 μL lysis buffer at 4 °C for 5 min. The supernatant was collected to determine the enzyme activity after centrifugation at 14,000 rpm for 1 min at 4 °C.

### 4.4. Circular Dichroism (CD)

IRES RNA oligonucleotides were dissolved in 50 mM Tris buffer (with or without 100 mM K^+^, pH 7.5) at a final concentration of 5 μM, heated at 95 °C for 10 min, and slowly cooled to room temperature. CD was detected with a J-815 CD spectrometer (Jasco International, Tokyo, Japan). The wavelength range of the spectrum collected was 220~350 nm, the step length of the light wave was 1 nm, and the response time was 1 s.

### 4.5. Polymerase Chain Reaction (PCR) and Quantitative Real Time PCR (qRT-PCR)

To test whether the insertion of IRES sequence led to mRNA splicing between RLuc and FLuc, PCR was used for sequencing the insertion detection. The upstream primer 5′-GCTAAGAAGTTCCCTAACACCGA-4′ was designed inside the *RLuc* gene region. The downstream primer 5′-GCGCTCGTTGTAAATGTCGTTAG-3′ was designed in the *FLuc* gene region.

In order to verify whether the insertion of IRES sequence affects the transcription of RLuc and FLuc mRNA, qRT-PCR was conducted. After the 293T cells were transfected with PR-IRES-F plasmid DNA for 24 h, the total RNA was extracted and reverse transcribed into cDNA. Then, qRT-PCR was performed for detection. Primers for *RLuc* were qF 5′-GATCACTACAAGTACCTCACCGC-3′, qR 5′-TGGTGCTCGTAGGAGTAGTGAAA-3′, for *FLuc* were qF 5′-TGGACATCACCTATGCCGAGT-3′, qR 5′-GTAAATGTCGTTAGCAGGGGC-3′. *GAPDH* was used as a referenced gene.

### 4.6. Western Blotting

To obtain the total protein, the treated cells were washed twice with PBS, lysed with RIPA (Beibo, Beijing, China) on ice for 30 min, and centrifuged for 10 min at 4 °C. The supernatant was collected to conduct Western blot analysis. The protein concentration was determined with a BCA Kit (Beibo, Beijing, China). Hence, 30 μg protein was separated by SDS-PAGE and transferred to the PVDF membrane. Incubation took place with anti-RBM4 antibody (1:2000, Proteintech, Wuhan, China) or anti-GAPDH antibody (1:5000) and the horseradish peroxidase (HRP) labeled secondary antibodies (1:10,000). Enhanced chemiluminescence (ECL) kit (Invitrogen, Carlsbad, CA, USA) was used to observe and record the experimental results.

### 4.7. Expression and Purification of Recombinant RBM4 Protein

RBM4 ORF was constructed into a pET-28a vector between the Nco I and Xho I restriction sites. The recombinant plasmid DNA was transformed into *Escherichia coli BL21* cells for protein expression. One mM IPTG (isopropyl-beta-d-thiogalactopyranoside) was used to induce the protein expression. The cells were collected after 4 h induction and re-suspended in 10 mL pre-cooled 20 mmol/L Tris-HCl buffer (containing 1 mol/L NaCl, 5 mmol/L β-Me, pH 8.0). After sonication and centrifugation, the supernatant was passed through a Ni-NTA nickel column (Thermo Scientific, Waltham, MA, USA) for purification. The nonspecific binding proteins were eluted with a gradient solution of imidazole prepared with buffer solution. The gradients of imidazole solution were 0, 20, 50, and 400 mmol/L, respectively. The elution of the target recombinant protein was validated by SDS-PAGE.

### 4.8. Electrophoretic Mobility Shift Assay (EMSA)

EMSA was conducted using LightShift^®^ Chemiluminescent EMSA Kit (Thermo Scientific, Waltham, MA, USA) according to the instruction. RNA probes (including 5′-FAM labeled sequences, 5′-FAM labeled mutated sequences and unlabeled sequences) were annealed to allow the formation of G4 structures (annealing method was based on CD experiment). The purified recombinant RBM4 protein (1 μg) was incubated with the annealed RNA G4 probes at room temperature for 20 min, and the reaction mixture was separated by 4% native PAGE electrophoresis. After electrophoresis, a gel imaging system was used to record the results.

### 4.9. Data Statistics

All data were statistically analyzed by GraphPad Prism8, shown as mean ± standard deviation, and pairwise comparison was performed with T test, where * *p* < 0.05, ** *p* < 0.01, *** *p* < 0.001 indicate statistically significant differences. All experiments were conducted for three repeats.

## Figures and Tables

**Figure 1 ijms-23-00743-f001:**
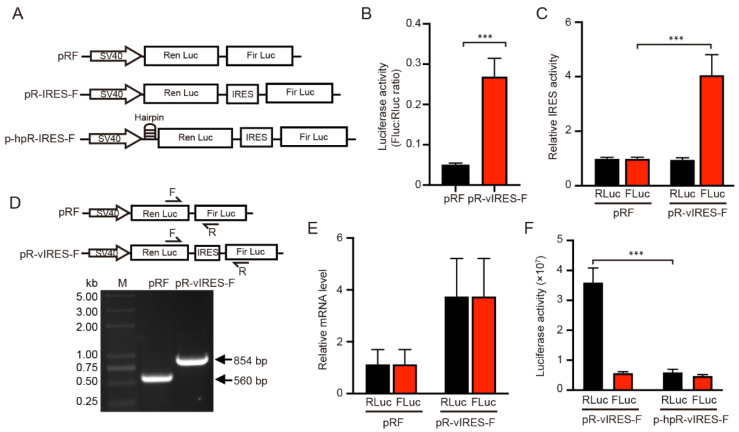
Dual luciferase assay for the functional analysis of *VEGFA* IRES-A (vIRES). (**A**) Schematic illustration of the dicistronic reporter gene vector constructs used in this study. (**B**,**C**) Influence of vIRES positioned upstream the firefly and renilla luciferases. The IRES activity are represented as ratios of firefly to renilla luciferase. The individual luciferase activity was normalized to that of the control vector pRF. (**D**) RT-PCR and (**E**) qRT-PCR verification of the influence of vIRES on the mRNA splicing and transcription. (**F**) Activity of the individual luciferases in the bicistronic vectors in the absence or presence of a hairpin upstream of renilla luciferase. *** indicates significant difference at *p* < 0.001.

**Figure 2 ijms-23-00743-f002:**
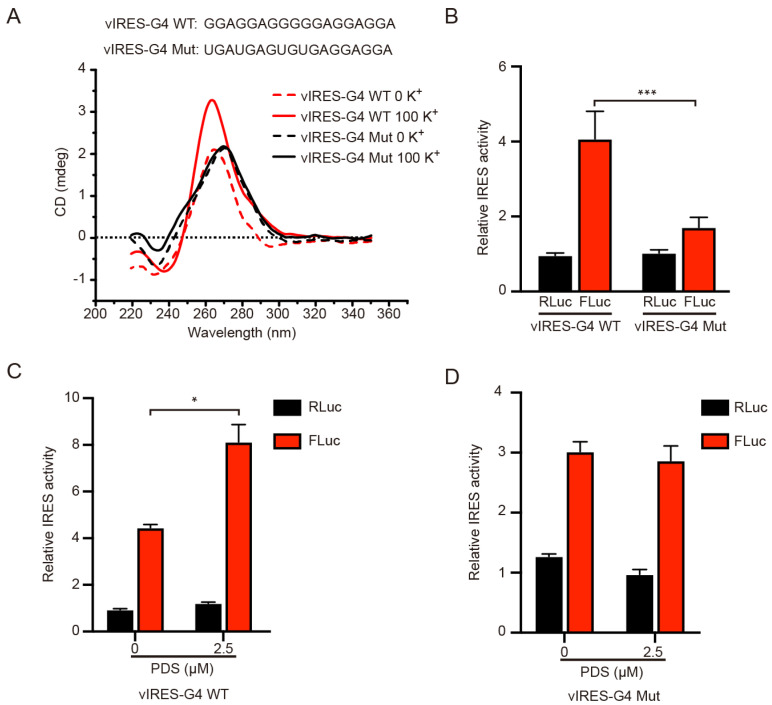
Effect of G4 structure on the individual luciferase activity. (**A**) CD analysis of the wild type and mutant G4 sequence in the vIRES vector in the absence or presence of 100 mM K^+^. (**B**) The influence of the G4 mutation on the FLuc luciferase activity. PDS treatment significantly enhanced the FLuc luciferase activity of the wild type G4 (**C**), but not the mutant G4 (**D**). * indicates significant difference at *p* < 0.05; *** indicates significant difference at *p* < 0.001.

**Figure 3 ijms-23-00743-f003:**
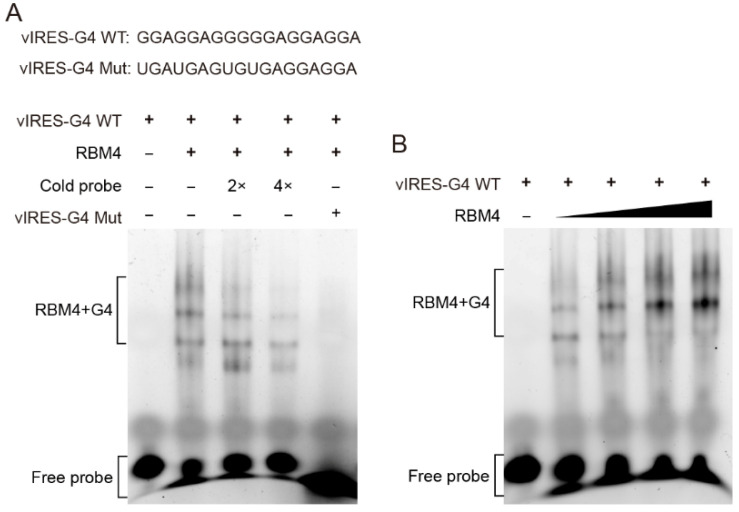
RBM4 specifically bound to the G4 structure in vIRES. (**A**) EMSA analysis of the binding between RBM4 and the G4 structure in the vIRES. (**B**) The binding between RBM4 and G4 was gradually enhanced with the increase of RBM4 protein level.

**Figure 4 ijms-23-00743-f004:**
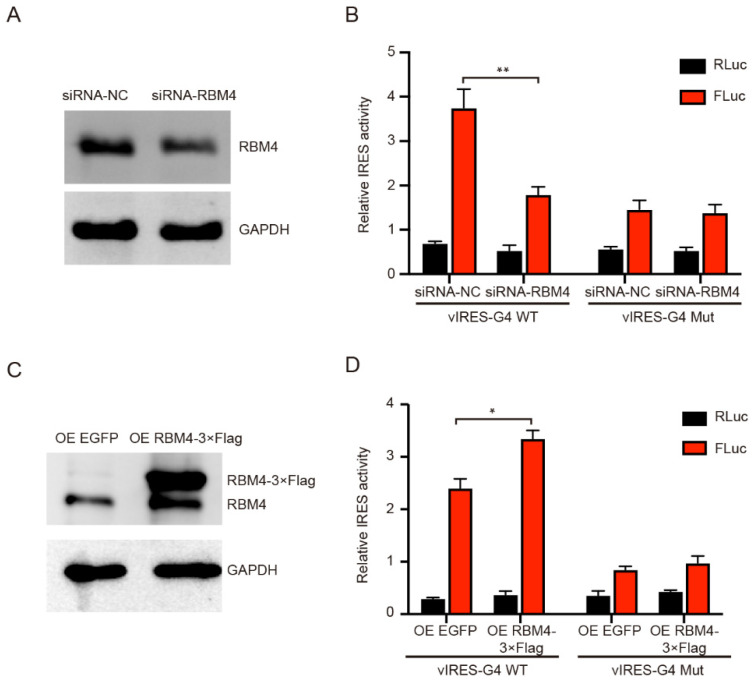
Effect of RBM4 on the expression of luciferases. (**A**) Western blot analysis of RBM4 protein after RNAi. (**B**) Knockdown of RBM4 inhibited the FLuc luciferase activity. (**C**) Western blot analysis of RBM4 protein after over-expression. (**D**) Over-expression of RBM4 resulted in the increase in the FLuc luciferase activity in the wild type IRES-A element, but not in the mutated element. * indicates significant difference at *p* < 0.05; ** indicates significant difference at *p* < 0.01.

**Figure 5 ijms-23-00743-f005:**
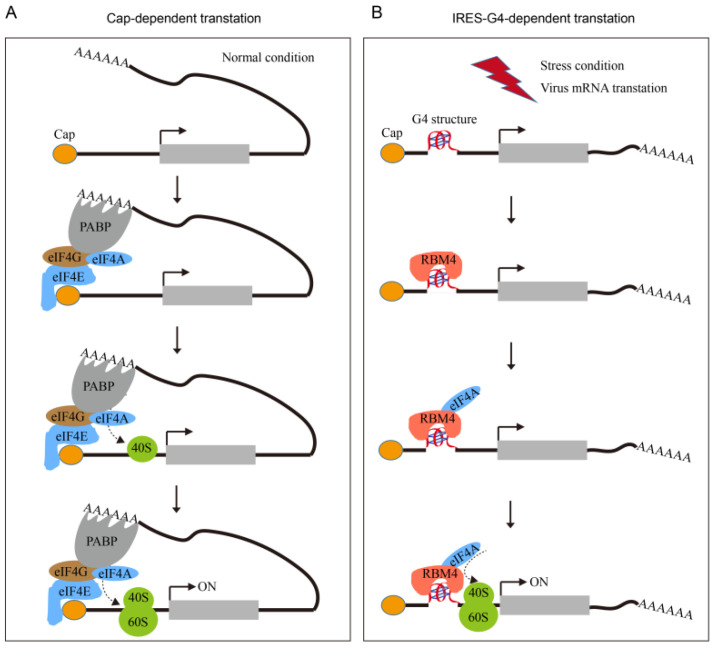
Diagram of proposed regulation mechanism of RBM4 and G4 structure on the translation of *VEGFA*. In normal conditions, *VEGFA* mRNA translates in a cap-dependent way (**A**). While, when cells are response to stress, *VEGFA* mRNA translates through IRES dependent way which needs the binding of RBM4 and G4 structure in the IRES-A element (**B**).

## Data Availability

All data generated or analyzed during this study are included in this published article.

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
