# Peer review of "G-Quadruplex Regulation of VEGFA mRNA Translation by RBM4"

_ijms, 2022, doi:10.3390/ijms23020743_

Round 1

Reviewer 1 Report

In the present study, Kangkang Niu, Xiaojuan Zhang et al. inspected the internal ribosome entry site (IRES) localized in the 5’-UTR of vascular endothelial growth factor A (VEGFA) mRNA. This IRES element is believed to contain a G-quadruplex forming locus which could regulate the activity of IRES (via G-quadruplex recognition by protein RBM4) and translation of VEGFA mRNA in general. This is, in fact, the main message of this study, authors brought a piece of solid evidence that protein RBM4 promotes the mRNA translation of the VEGFA gene by binding to the G-quadruplex in the IRES. Altogether, the paper is well written, and experiments are adequately chosen and complementary to each other. Nonetheless, I would recommend adding a simple schematical figure - something like "take-home message" - in order to illustrate the main idea of the paper to the wide readership of the IJMS journal (can be easily done e.g. in the free version of the Biorender tool: https://biorender.com/). In addition, I have some major and minor points on how to improve the paper:

Major point:

  • Do authors think that this particular type of VEGFA regulation is evolutionarily conserved? In other words, is G-quadruplex forming sequence in IRES-A conserved at least in mammals?  Multiple sequence alignment of IRES in a few representative species would greatly strengthen the significance of the paper. Could authors add a short paragraph dealing with this?
  • Authors stated: "Moreover, RBM4 known as an RNA binding protein here was verified to be involved in cap-independent translation as G4 binding protein. This may provide a novel potential therapeutic target to affect VEGF-A expression in tumors." I believe that this statement about a novel potential therapeutic target should be discussed more, authors should provide some additional clues to stimulate further research in this field.
  • The authors should briefly discuss the binding mode of RBM4 to G-quadruplex - do they think it functions as a G4 stabilizer or destabilizer? Because basically some G4 proteins stabilize G4s and some of them are resolving them. This should be explicitly stated.

Minor points:

  • line 56 ... "2 to 4 G planes are stacked to form G4 structure" ... this is not true, as G4s with more than 4 G planes can be formed (see e.g. https://pubmed.ncbi.nlm.nih.gov/29733879/)
  • line 71 ... medicated should be mediated (?)
  • line 74 ... Bombyx mori ... Latin names should be in Italics (also line 157 in vitro, Escherichia coli, gene names, etc.)
  • VEGFA, VEGF-A, VEGF-a should be unified
  • line 223 ... highly homologous ... please add exact value (percentage)
  • line 250 ... typo in "was synthesizd and"
  • line 287 ... "IRES RNA oligonucleotides were dissolved in 50 mM Tris buffer (pH 7.5)" ... what type of ions were used to stabilize G4?
  • line 311 ... proteintech
  • line 312 ... abbreviation ECL should be explained

Reviewer 2 Report

The article “G-quadruplex regulation of VEGFA mRNA translation by RBM4” by Niu et al. describes that RBM4 binds the VEGFA mRNA G4 structure, and promotes the translation activity of the IRES-A element of the mRNA by using cell-based assay. The topic is important, and the manuscript is well written. However, I wish the authors address the following comments. It should be necessary to show additional experiments. The data representation of the results must be improved.

Comments:
(1)
In Figure 2a, it appears that there are G4 signals in both WT and Mut, so add the results of CD measurements using desalted samples of both WT and Mut. Moreover, add the concentration information of the cation to the experimental section.

(2)
In Figure 3, where is the G4-only band? Is the band labeled "free probe" G4? If so, where is the probe in single-stranded state (unstructured)? Please add your comments.
Moreover, in Figure 3B, there are binding bands that become weaker when the concentration of RBM4 is increased, what is happening here? Please add your comments.

Minor comment:
Some typo in the manuscript should be corrected. Please check all typos again in your manuscript.
e.g.
Line 157, “in vitro” should be corrected to “in vitro” (italic).
Line 235, “VEGF-a” should be corrected to “VEGF-A”.

Round 2

Reviewer 2 Report

I have checked the revised manuscript. I am satisfied with the response of the authors and the revisions. Therefore, I recommend acceptance of this manuscript.